# Removal of Cylindrospermopsin by Adsorption on Granular Activated Carbon, Selection of Carbons and Estimated Fixed-Bed Breakthrough

Caio César Antonieti and Yovanka Pérez Ginoris *

Department of Civil and Environmental Engineering, University of Brasília, Brasília 70910-900, Brazil; caioc.antonieti@hotmail.com
* Correspondence: yovanka.perez@gmail.com

**Abstract:** Climate change and the increase in the availability of nutrients in aquatic environments have increased the occurrence of cyanobacterial blooms which can produce cyanotoxins such as cylindrospermopsin (CYN). Activated carbon adsorption have been proved to be efficient for CYN removal. In the present study, a carbon with high CYN adsorption capacity was identified between two granular activated carbons. For this carbon was estimated the operating time of a full-scale granular activated carbon column under different empty bed contact times (EBCT). The fixed-bed breakthrough was estimated using the Homogeneous Surface Diffusion Model (HSDM). Wood carbon showed greater capacity to remove CYN. The experimental equilibrium data best fitted Langmuir isotherm model, in which wood carbon had a maximum adsorption capacity of 3.67 µg/mg and Langmuir adsorption constant of 0.2791 L/µg. The methodology produced satisfactory results where the HSDM simulated the fixed-bed breakthrough with a coefficient of determination of 0.89, to the film diffusion coefficient ($K_f$) of $9 \times 10^{-6}$ m/s and surface diffusion coefficient ($D_s$) of $3 \times 10^{-16}$ m$^2$/s. It was observed that the increase in EBCT promotes a reduction in the carbon use rate. The best carbon use rate found was 0.43 kg/m$^3$ for a EBCT of 10 min and breakthrough time of 183.6 h.

**Keywords:** adsorption; granular activated carbon (GAC); cylindrospermopsin (CYN); Homogeneous Surface Diffusion Model (HSDM); SBA test; breakthrough curves

## 1. Introduction

Global warming and the increasing availability of nutrients in aquatic environments have contributed to the increase in the occurrence of cyanobacterial blooms [1,2]. Cyanobacteria constitute a severe environmental and public health problem, as they can produce toxic metabolites called cyanotoxins. Contact with or ingestion of water containing cyanotoxins can cause serious harm to human and animal health, including damage to the digestive, endocrine, dermal and nervous systems [3,4].

Cylindrospermopsin (CYN) is a cyanotoxin produced by several cyanobacterial genera. CYN has hepatotoxic, general cytotoxic and neurotoxic effects [5]. In mammals, CYN intoxication causes damage to the liver, kidney, thymus, and heart [6,7]. In vitro assays also suggest that CYN has the potential to act as an endocrine disruptor by altering the progesterone/estrogen ratio in women [8].

In this case, species of cyanobacteria were identified as producing CYN, with the record of the occurrence in countries in Europe, North America, South America, Central America, Asia and Oceania [9].

The World Health Organization [10] presents provisional values of guidelines for consumption and recreational activities in waters containing CYN. The established values of CYN concentrations are: 0.72 µg/L for water intake for long periods, 3.0 µg/L for water intake for short periods and 6.0 µg/L in water intended for recreational activities.

The Brazilian Ministry of Health [11], through Ordinance GM/MS 888 of 4 May 2021, establishes the concentration of 1 µg/L as the maximum allowed value of CYN in water for human consumption.

CYN is an alkaloid that contains a tricyclic guanidine group combined with a hydroxymethyluracil (Figure 1) [5]. Two naturally occurring analogues of the molecule were identified, namely: 7-epi-CYN [12], 7-deoxy-CYN [13]. However, it has not been fully elucidated whether the 7-deoxydesulfo-CYN and 7-deoxydesulfo-12-acetyl-CYN analogues are congeners or degradation products of the molecule [14].

**Figure 1.** Molecular structure of common cylindrospermopsins. [10].

CYN is relatively stable in the dark with a slow degradation at temperatures above 50 °C; however, in the presence of sunlight and photosynthetic pigments, degradation can occur rapidly leading to the destruction of 90% of the total toxin within two to three days [4]. Its molecular mass is 415.43 Da, its pKa is 8.8. The structure of the molecule gives it high polarity, so it has high solubility in water.

The CYN-producing cyanobacterial genera can produce the toxin intra and extracellular, with the extracellular concentration tending to be significantly higher than the intracellular one, constituting up to 90% of the total available toxin [15,16]. The high extracellular concentration makes CYN especially important, as there is no need for cell lysis for the water to be contaminated.

It is widely reported in the literature that water treatment technologies involving chemical coagulation, including conventional treatment, are not effective in removing the extracellular cyanotoxin fraction, despite being efficient in removing cells carrying the intracellular fraction [17–20].

Chlorine oxidation allows the degradation of dissolved CYN [21,22]; however, according to review studies of Merel et al., [23], it was identified that the use of chlorine in the oxidation of CYN caused the formation of two by-products whose toxicity should be studied. In this sense, technologies that allow the removal of dissolved cyanotoxin and that do not cause the formation of by-products are necessary in water treatment plants whose supply sources present occurrences of CYN-producing cyanobacterial blooms.

Several studies demonstrate that adsorption on activated carbon is efficient in removing dissolved cyanotoxins without generating by-products [24–27].

Activated carbon is a carbonaceous material that can be produced from a wide variety of raw materials. Usually, carbonaceous material is mixed with dehydrating and oxidizing chemicals and then heated to 400–800 °C in the absence of air. As activating agents, potassium hydroxide, zinc chloride or phosphoric acid can be used. In Brazil, wood, bituminous and sub-bituminous carbon, bone and coconut shell are predominantly synthetized and used in water treatment plants to remove organic micropollutants.

Depending on the size of the particles that make up the carbon, this adsorbent is available in pulverized (PAC) and granular (GAC) forms. In general, the use of PAC requires less expensive installations, and its application can be seasonal. However, its use implies an increase in sludge production in the treatment plant and the impossibility of its reuse. In contrast with PAC, GAC can be regenerated and represents a constant barrier to the presence of contaminants in the supply source [28].

The GAC application usually occurs in columns or filters that support the material in a steady-state and are subjected to a water flow containing the target pollutant.

It is essential to know the fixed-bed breakthrough to assess the technical and financial feasibility of choosing the use of GAC in a treatment system. The fixed-bed breakthrough depends on the efficiency of the adsorptive process. In turn, the adsorption process depends on the adsorbent's specific surface, pore size and structure, reactivity between the different components of the material, characteristics of the adsorbate (size, molar mass, and functional groups of molecules), and water characteristics (natural organic matter, turbidity, pH, and temperature), among other factors. According to Di Bernardo et al., [29] there is a relation between the molecular mass of the target pollutant and the activated carbon pore size, making each activated carbon specific to remove a particular pollutant. Accordingly, it is essential to have prior knowledge of activated carbon's main properties to be applied and to carry out studies aimed at removing the target pollutant for the choosing the carbon that has the highest adsorption capacity and the feasibility of its use is verified.

Column performance and the time required to replace the GAC can be estimated through tests in a pilot installation, mathematical models, and rapid small-scale column tests (RSSCT).

Testing in a pilot installation is the most reliable way to represent the performance of a GAC column [30]; however the time and financial costs for carrying out the studies are high and, as they demand large volumes of water, is impossible its execution in the laboratory.

Mathematical models have been widely used to estimate the dynamics of the operation of a GAC column. Specifically, Homogeneous Surface Diffusion Model (HSDM), which has been applied in studies with several pollutants providing reliable information that reproduces the operation of a column of Full-scale [31–36]. The model considers the adsorption process occurring in two sequential diffusion steps, namely: diffusion through the stationary fluid layer surrounding the adsorbent particle (film diffusion) and diffusion through the internal pores of the adsorbent (surface diffusion).

However, in order to obtain reliable simulations, the precise determination of the model's input parameters is essential. Applying the HSDM model requires accurately determining the adsorption equilibrium parameters and the mass transfer coefficients (film diffusion coefficient and surface diffusion coefficient) [37]. Equilibrium parameters are commonly estimated by determining the adsorption isotherms [38]. Batch kinetic studies or column tests, among other approaches, can be applied to determine the mass transfer coefficients.

Weber and Liu [39] developed the methodology of short bed adsorber tests (SBA) which allows the determination of the mass transfer coefficients simultaneously. The methodology consists of operating a system of fixed bed columns that operate with the same granulometry and surface flow rate as the full-scale columns; however, these columns have a sufficiently reduced bed thickness to obtain immediate rupture in their operation. Thus, the initial concentration influent to the column is different from zero [39,40].

Comparing the breakthrough obtained by the SBA test and the breakthrough provided by the HSDM model make it is possible to obtain the mass transfer coefficients.

Several works demonstrate the effectiveness of SBA tests in estimating mass transfer coefficients for various contaminants [31–33,40]. While the SBA constitutes a methodology that allows obtaining the mass transfer coefficients necessary for the application of HSDM model, the RSSCT is a test that can produce the same fixed-bed breakthrough of a full-scale GAC column, dispensing with the application of mathematical models, via bench-scale tests using GAC granulometry and axial velocity different from those used in a full-scale column. However, using RSSCT is conditioned to a previous validation of the method for the adsorbents and adsorbates studied [41].

Although studies prove the efficiency of activated carbon adsorption technology to remove dissolved cyanotoxins, studies evaluating fixed-bed adsorption columns employing GAC for CYN removal are rare or non-existent. In this context, using an ultrafiltered base water, the present work aimed to: (1) identify, between two commercial GACs of different

origins, the one with the highest CYN adsorption capacity. For the highest capacity GAC, (2) obtain the adsorption equilibrium parameters and the mass transfer coefficients (film diffusion coefficient and surface diffusion coefficient) necessary to apply the HSDM model (3) apply the HSDM model to estimate the Fixed-bed breakthrough of a full-scale granular activated carbon column under different EBCT.

## 2. Materials and Methods

The study included two experimental stages. Initially, the carbons were characterized and then tests of adsorptive capacity were carried out to choose the carbon with the highest capacity to remove CYN. Subsequently, short bed adsorber tests were carried out, followed by mathematical modeling to estimate the Fixed-bed breakthrough of a full-scale granular activated carbon column. The Paranoá Lake water, after ultrafiltration, was used as the base water in all the experimental stages.

### 2.1. Study Water

The base water was collected at the Water Treatment Plant (WTP), Lago Norte, located in Brasilia, Federal District, Brazil. Paranoá Lake is the source supply of the WTP that possesses ultrafiltration technology in its treatment process. The base water was collected after the ultrafiltration step that precedes the chlorination step and characterized in relation to pH and turbidity, measurements of total organic carbon were not performed in this study.

Prior to carrying out the tests intended for each experimental stage, the base water was collected for the preparation of the study water. The preparation of the study water consisted of adding a CYN extract to the collected base water. The CYN extract was obtained from the lysis of the cultured strain of CYN-producing cianobacteria Raphidiopsis Raciborskii, formerly called Cylindrospermopsis raciborskii.

The cultivation of Raphidiopsis raciborskii was carried out at the Environmental Sanitation Laboratory of the University of Brasília. The cultured cells were lysed through ice and thawing steps three consecutive times. Subsequently, the lysed culture was filtered using filter paper, then filtration on a fiberglass membrane and finally filtration on a 0.45 μm cellulose ester membrane.

The lysed extract was added to the base water aiming at an initial concentration of CYN in the study water of 100 μg/L. The CYN concentration of 100 μg/L was adopted because it was the maximum concentration found in monitoring of raw water reservoirs in Australia and the United States of America [42,43].

Before the beginning of each test, the pH of the study water was adjusted to a value of 6.5, using 0.5 M HCl or 0.25 M NaOH solutions. The slightly acidic pH of the study simulated the pH of the ultrafiltrated water prior to the disinfection step. After preparing the study water, the initial concentration of CYN was quantified by Enzyme-Linked Immunosorbent Assay (ELISA) test kits produced by EUROFINS ABRAXIS.

### 2.2. Characteristics of Activated Carbons and Adsorption Equilibrium Tests

Brazilian manufacturers kindly supplied the evaluated carbons. For the development of the studies, the activated carbons were characterized concerning the BET surface area, pore volume distribution, and the point of zero charge (pHpzc). The manufacturers specified the raw material used to produce each carbon when supplying the materials, with a carbon produced from wood and the other from bitumen.

The BET surface area and the volumes of micro and mesopores were determined using nitrogen adsorption isotherms at −196 °C, obtained in the Quantachrome Nova 2200 equipment (Quantachrome Instruments, Boynton Beach, FL, USA).

Once the adsorption isotherms were obtained, the surface area and volume of micropores were determined by applying the Brunauer–Emmett–Teller and Dubinin–Radushkevich equations. It was considered that the sum of the volume of micro and mesopores of the carbon samples corresponds to the volume of liquid nitrogen adsorbed at a relative pressure

of 0.95 ($V_{0.95}$). Thus, the volume of mesopores was obtained by subtracting the volume of micropores from $V_{0.95}$.

Additionally, the carbons were characterized concerning the point of zero charge ($pH_{pcz}$), according to the methodology proposed by Moreno-Castilla et al., [44].

The adsorption equilibrium tests were performed to determine the equilibrium time for each carbon, according to the D3860-98 standard [45]. In its procedures, the standard indicates the use of a contact time between the carbon and the solution containing the pollutant of two hours, considering that this time is sufficient to stablish the equilibrium of the adsorption process. However, the standard indicates the need to conduct studies to verify if this time is sufficient to reach the equilibrium condition. Thus, previous to the adsorption equilibrium assays, tests were carried out to define the time required to reach the adsorption process equilibrium.

For the adsorption tests, suspensions were prepared with each of the GACs. For the preparation of the suspensions, the GACs were submitted to grounding until 95% of the mass of the carbon samples passed through a sieve of 325 mesh (0.044 mm); then the carbons with reduced granulometry were dried in an oven at 150 °C for 3 h. After that, the carbons were stored in a desiccator for cooling until to reach the room temperature. The dried carbons were weighed, and suspensions were prepared by adding ultrapure water free of $CO_2$. The suspensions remained for another 12 h in a vacuum desiccator (Nalgon, Itupeva, Brazil) under a negative pressure of 600 mmHg to eliminate the air present in the carbon interstices [20].

The adsorption tests to determine the equilibrium time were carried out in duplicate for each of activated carbon. Seven 1 L beakers with 500 mL of study water were used. Each beaker received an aliquot of activated carbon suspension to obtain the initial concentration of 80 mg/L in the water study. The beakers were placed in a jar testing equipment, maintaining the mixing speed at 205 r.p.m [46] in an environment with a constant temperature of 23 °C. The contact times adopted were: 0.5, 1, 2, 3, 6, 12, and 24 h. After the contact time, each beaker was removed from the agitation, the study water was immediately filtered through a cellulose ester membrane with a pore size of 0.22 μm, to separate the carbon and interrupt of the adsorptive process. The filtered water fractions were analysed for pH value and CYN concentration. Figure 2 present a schematic diagram of the conditions evaluated in the assays of equilibrium time determination.

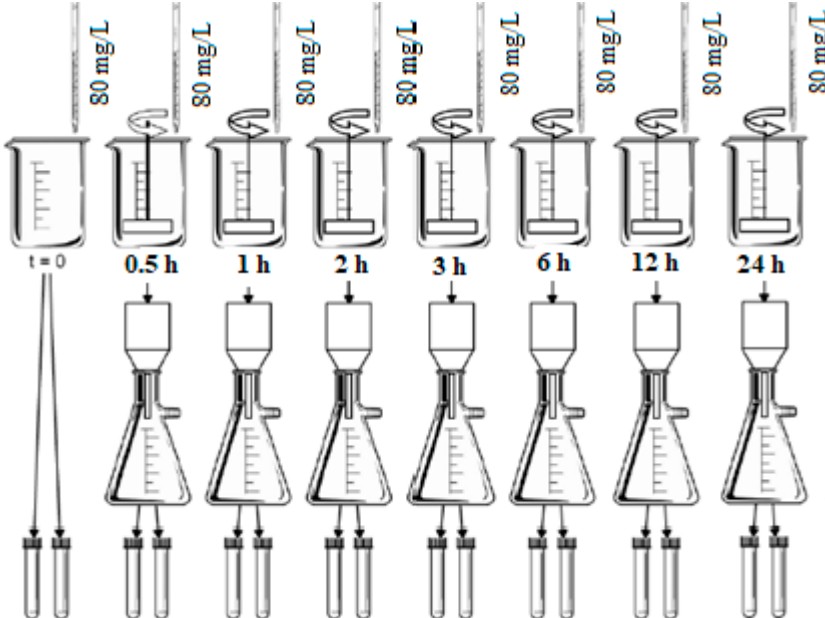

**Figure 2.** Schematic diagram of the equilibrium time test.

The experimental data obtained in the equilibrium time tests allowed to define, for each carbon, the time of adsorption equilibrium to be used in the subsequent adsorption equilibrium tests. The procedures followed in the adsorption equilibrium tests were the same adopted in the in the adsorption tests to determine the adsorption equilibrium time, varying only the concentration of carbon in each beaker and the contact time of each the carbons studied. The concentrations of the carbon suspension evaluated in the adsorptive capacity tests were: 0, 10, 20, 30, 40 and 70 mg/L for both carbons. The agitation time for the tests performed with each activated carbon corresponded to the equilibrium time defined in the equilibrium time tests. After the contact time, the water treated in each experimental condition was filtered and analysed in relation to the pH value and the CYN concentration. The adsorption equilibrium tests were performed in duplicate for each studied carbon.

With the experimental data, the mass of CYN adsorbed by each carbon was calculated using the Equation (1):

$$q = \frac{(C_0 - C_e)V}{m} \tag{1}$$

where:

q: Adsorption capacity ($\mu$g/mg)
$C_0$ : Initial CYN concentration in water ($\mu$g/L)
$C_e$: Aqueous phase CYN concentration at equilibrium ($\mu$g/L)
V: Volume of the liquid phase (L)
m: mass of the dosed activated carbon (mg)

### 2.3. Short-Bed Adsorber Test (SBA)

Short-Bed Adsorber tests (SBA) were performed to evaluate the kinetics of CYN adsorption in fixed bed adsorbent columns [39,40]. The assays were carried out in duplicate with the GAC that showed the highest adsorption capacity. Table 1 presents the operational conditions used in the SBA tests.

**Table 1.** SBA operating conditions.

| Conditions | Unit | GAC 18 × 25 Mesh |
|---|---|---|
| GAC bed height | cm | 3.0 |
| Average particle diameter | mm | 0.855 |
| Column diameter | mm | 10 |
| Axial velocity | $m^3/(m^2.day)$ | 200 |
| Flow Rate | L/h | 0.6542 |
| EBCT | s | 13.06 |
| Initial CYN concentration | $\mu$g/L | 109.25 |
| Apparent (bed) density | $g/cm^3$ | 0.476 |
| Bed porosity | - | 0.44 |
| Adsorbent mass | g | 1.1296 |

The GAC was crushed and sieved to produce the defined granulometric fraction (Table 1). After grinding and sieving, the carbon was carefully washed with ultrapure water to remove all the dust and the pH of the wash water did not change significantly. The washed GAC was dried in an oven at 150 °C for 4 h, then, it was placed in a desiccator until reaching room temperature. After cooling, the apparent density of the carbons was determined, according to the technical standard ASTM D 2854 [47]. Before being placed in the column, the carbon was placed in 40 mL of ultrapure water and heated, remaining at boiling point for 10 min to remove air from the pores. After cooling to room temperature, the GAC suspension was carefully inserted into the column [48].

The experimental system for carrying out the SBA test was composed of two 30 L polyethylene water tanks, one for storing the study water and the other for collecting the treated effluent. The piping used was low-density polyethylene, a column of Polytetrafluoroethylene (PTFE), connections, and valves of galvanized steel. A peristaltic pump with flow adjustment was used to ensure the flowrate established for the tests. A manometer was installed upstream of the column for pressure monitoring.

Figure 3 shows the system used in SBA tests. The GAC bed was fixed to the column using a 100 Mesh stainless steel mesh (0.149 mm), fixed through a sealing ring. The column was kept flooded by positioning the effluent outlet tube higher than the activated carbon inside the column. The system was projected so that the bed volume of the adsorbent would permit the ratio of Bed Height/Particle Diameter higher than 20, thus, disregarding the axial dispersion [31]. The Inner column diameter/Particle diameter ratio was higher than 8 so that wall effects in the mass transfer process could also be disregarded [49].

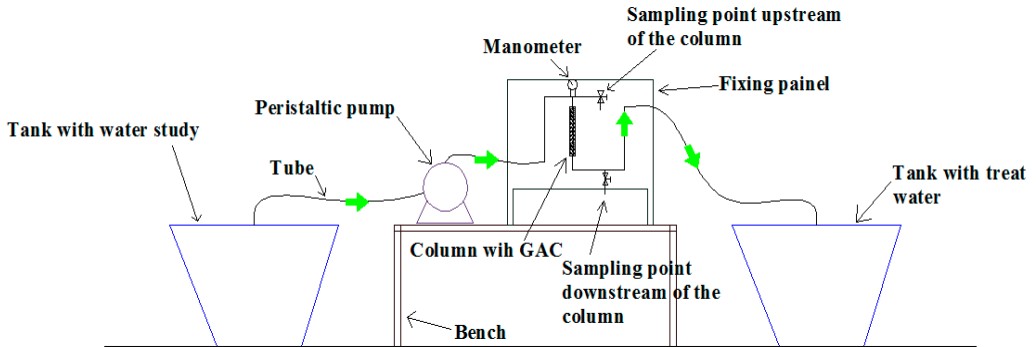

**Figure 3.** Experimental system used in SBA test.

Kamp et al. [50] studied the adsorption of microcystin, cylindrospermopsin and saxitoxin by the following materials: glass, high-density polyethylene, polyethylene glycol terephthalate, polycarbonate, polypropylene, and polystyrene. The authors found that CYN is not adsorbed by any of these materials. However, the researcher did not study the materials used in the present study. Thus, preliminarily, an assay was carried out using glass microspheres replacing the GAC bed to evaluate the adsorption of CYN in the materials that make up the system.

Before the beginning of each test, the system with GAC was operated for 1 h using ultrapure water to eliminate possible air bubbles in the GAC.

CYN concentration was monitored in the influent and the effluent to the SBA column by analysing samples collected at the following test times: 0, 15, 30, 60, 120, 600, and 900 min.

### 2.4. Modeling the Adsorption Process

The experimental data from the adsorption equilibrium tests obtained in the first experimental stage were fitted to the isotherm models Langmuir, Freundlich and Redlich-Peterson. The parameter values of the isotherm model that presented the best fit to the adsorption equilibrium data were applied to the HSDM model to simulate the adsorption process in SBA columns.

The model considers the adsorption process occurring in two sequential diffusion steps, namely: diffusion through the stationary fluid layer surrounding the adsorbent particle (film diffusion) and diffusion through the internal pores of the adsorbent (surface diffusion) as depicted in Figure 4.

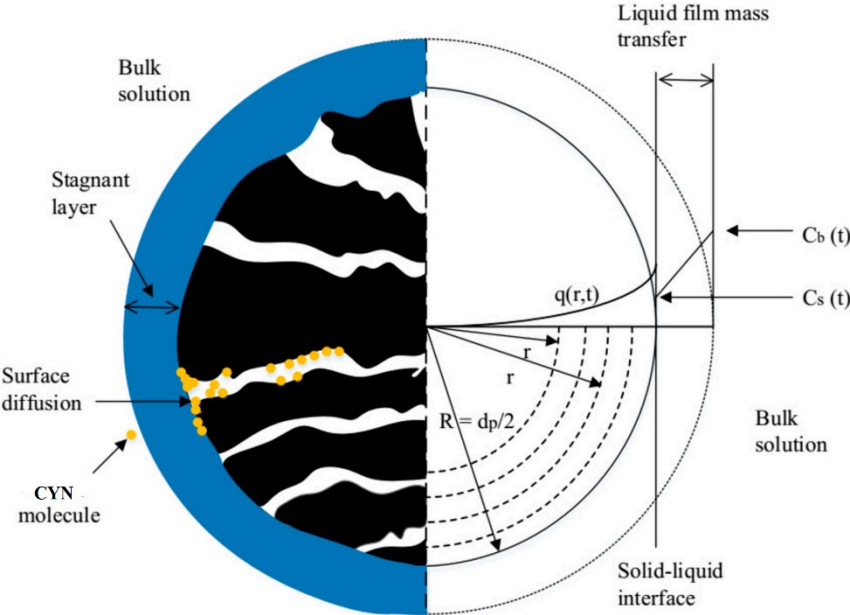

**Figure 4.** Illustration of the mass transfer–adsorption process into a particle of GAC.

Adapted from Usman et al., 2021 [35]

The following conditions are considered in the model:

1- The dispersion of the pollutant through the bed occurs according to a plug-flow system
2- The applied hydraulic load is constant.
3- Internal diffusion is the predominant mechanism in mass transport and is independent of the pollutant concentration.
4- The diffusion in the film is carried out by a linear driving force.
5- The adsorbent particles are spherical and homogeneous.

The equations that describe the HSDM are presented in Table 2.

**Table 2.** Equation for the Homogeneous Surface Diffusion Model. Adapted from Roy et al., [51].

| Equation | Role |
|---|---|
| $v\left[\frac{dC}{dZ}\right]_t + \left[\frac{dC}{dZ}\right]_z + \rho_P\left(\frac{1-\varepsilon}{\varepsilon}\right)\left[\frac{dq}{dt}\right]_z = 0$ <br> Where: <br> C: Adsorbate concentration in the liquid phase at time t $(ML^{-3})$ <br> $C_o$: Initial concentration (in the influent) of adsorbate in the liquid phase $(ML^{-3})$ <br> t: Time (T) <br> Z: Bed height (L) <br> q : Adsortvate concentration in the solid phase $(MM^{-1})$ <br> $\rho_p$ : Apparent density $(ML^{-3})$ <br> $\varepsilon$: Bed porosity (dimensionless) <br> Initial conditions and limits: <br> t = 0,　$0 \leq Z < L$, C = 0 <br> t > 0,　Z = 0, C = $C_o$ | Fixed bed mass balance considering the plug-flow reactor hydraulic model |
| $\rho_P\left[\frac{dq}{dt}\right]_z = \frac{3K_f}{R_P}(C - C_s)$ <br> Where: <br> $R_p$ : Adsorbent particle radius (L) <br> $K_f$: Film diffusion coefficient $(LT^{-1})$ <br> $C_s$ : Adsorbate concentration in the liquid phase at the solid-liquid interface $(ML^{-3})$ | Linear driving force |

**Table 2.** *Cont.*

| Equation | Role |
|---|---|
| $\frac{dq}{dt} = \frac{D_s}{r^2}\frac{d}{dr}\left(r^2\frac{dq}{dr}\right)$<br>Where:<br>Ds: Surface diffusion coefficient ($L^2 T^{-1}$)<br>r : Radial coordinate (L) | Diffusion equation for a spherical particles |
| $q(r,0) = 0$ | Initial condition |
| $\frac{dq}{dr} = 0$ for r = 0 | Boundary condition for the center of the spherical particle |
| $\rho_p\, D_s\frac{dq}{dr} = K_f(C - C_s)$ | Boundary condition for continuity of flux<br>at r = $R_p$ |

Calculations in the HSDM model were performed using the FAST 2.1 software [52]. By maximizing the coefficient of determination between the experimental data and the data predicted by the model, it was possible to determine the mass transfer coefficients $K_f$ (film mass transfer) and $D_s$ (surface diffusion).

The values of the adsorption equilibrium parameters and the $K_f$ and $D_s$ coefficients showing the best representation of the breakthrough curve in the SBA tests were used in the HSDM model to simulate the behavior of a full-scale fixed bed adsorbent column applying different empty bed contact times.

## 3. Results and Discussion

### 3.1. Characteristics of Activated Carbons

Figure 5 presents the $N_2$ adsorption/desorption isotherms obtained from the textural analysis of activated carbons.

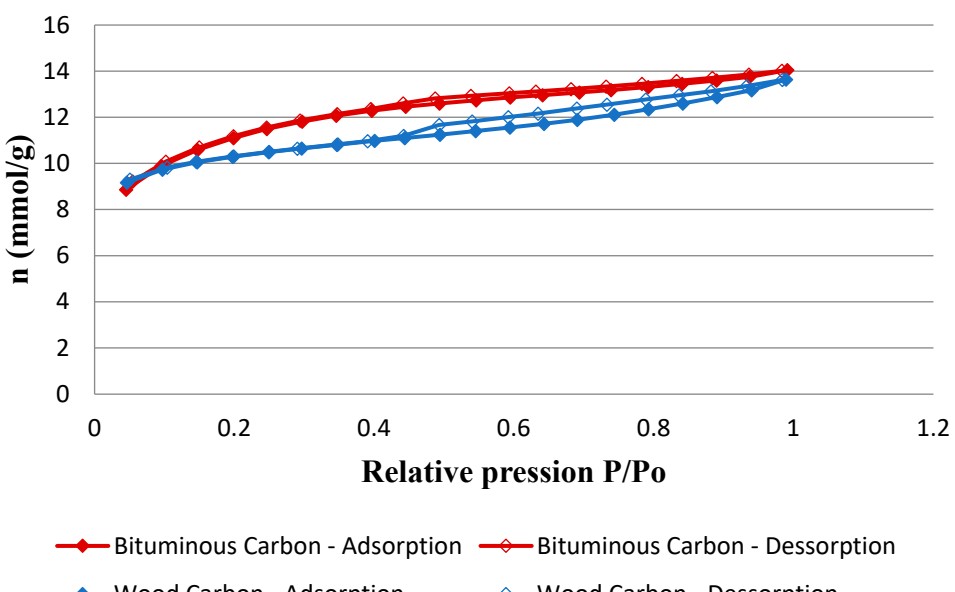

**Figure 5.** $N_2$ adsorption isotherms for the studied carbons.

Both carbons have type IV-A isotherms, which are typical of mesoporous adsorbents according to the IUPAC classification [53]. Wood carbon showed an isotherm with a hysteresis loop more pronounced than the hysteresis loop shown by bituminous carbon. This behavior indicates a higher volume of mesopores of the wood carbon. Moreover, wood carbon adsorbs a smaller volume of gas at low relative pressures, indicating that this carbon has a smaller volume of micropores when compared to bituminous carbon.

Table 3 presents the BET specific area values, the pore volumes obtained from the adsorption/desorption isotherms, as well as the zero charge point for each studied carbon.

**Table 3.** Physical characteristics of the studied carbons.

| Feature | Wood Carbon | Bituminous Carbon |
|---|---|---|
| Material | Wood | Bitumen |
| Area BET ($m^2/g$) | 726 | 819 |
| Micropore Volume ($cm^3/g$) | 0.375 | 0.423 |
| Mesopore Volume ($cm^3/g$) | 0.081 | 0.053 |
| $V_{0.95}$ ($cm^3/g$) | 0.456 | 0.476 |
| $pH_{pzc}$ | 8.61 | 8.49 |

The results of textural analysis, related to the pore volume, corroborate the characteristics of the adsorption isotherms exhibited by each of the carbons. Wood carbon contains a mesopore volume 35% higher and a 13% lower micropore volume than bituminous carbon. Wood carbon's BET area is lower than bituminous carbon, indicating the higher micropores volume of the latter carbon. The zero charge points of both carbons are numerically close, which indicates that both carbons must have similar surface electrical properties when applied in an aqueous medium with the same pH.

### 3.2. Adsorption Tests to Define the Equilibrium Time

Figure 6 illustrate the average values of CYN (C)'s residual concentration fraction ($C/C_0$) as a function of time for bituminous and wood carbon, respectively.

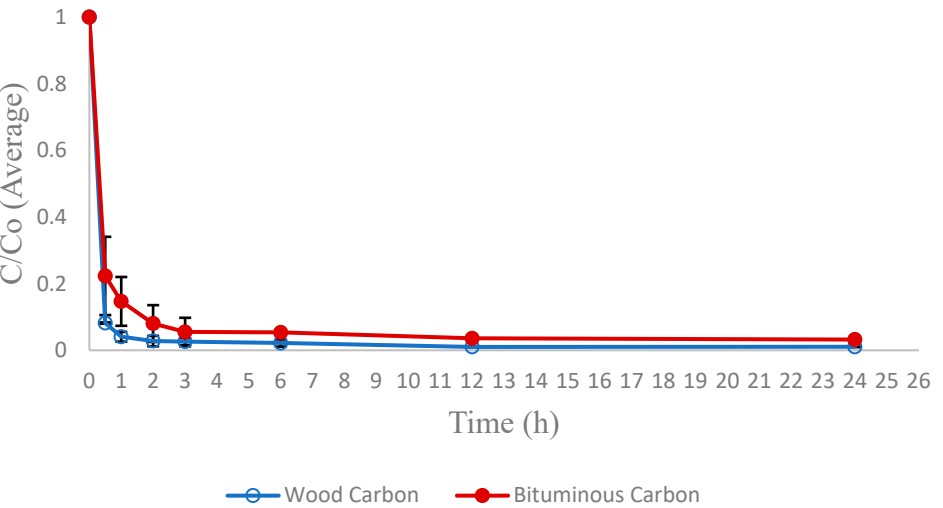

**Figure 6.** Results of equilibrium time tests performed for Bituminous and Wood GAC. C0 = Initial CYN concentration; C = CYN concentration in water after the contact time; initial pH of the study water = 6.5; Carbon dose = 80 mg/L; Base water (test with Bituminous GAC): pH = 7.5; Turbidity = 0.05 uT; Base water (test with Wood GAC): pH = 7.6; Turbidity = 0.07 uT.

Both carbons presented CYN removal higher than 80% in the first hour of contact and reached the adsorption equilibrium within 12 h of contact. Thus, this time was the contact time in the adsorption equilibrium tests.

### 3.3. Adsorption Equilibrium Assays

Figure 7 shows the percentages of CYN removal as a function of carbon concentration for wood and bituminous carbon.

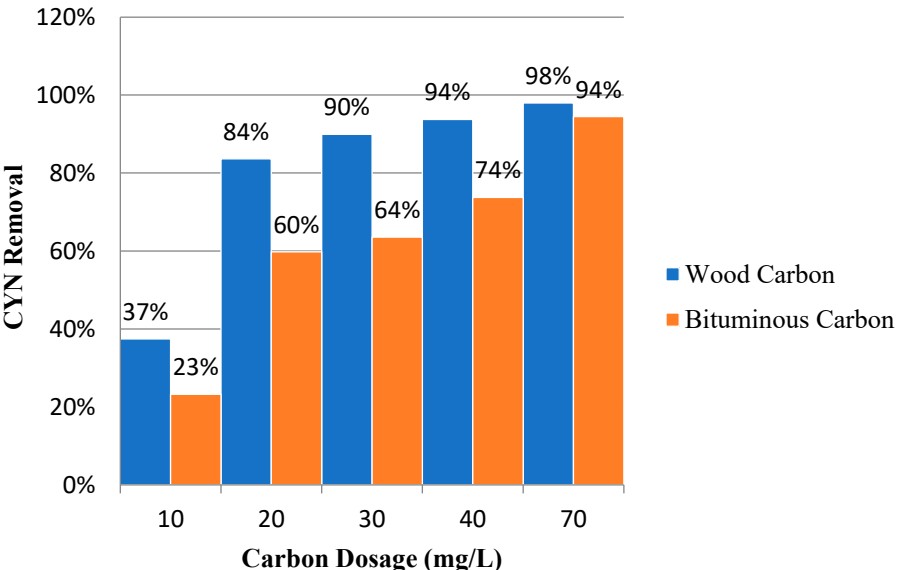

**Figure 7.** CYN removal as a function of the applied carbon concentration in Paranoá lake water matrix (C0—Bituminous Carbon = 102.23 μg/L; C0—Wood Carbon = 89.24 μg/L); initial pH of the study water = 6.5. Base water: pH = 7.7; Turbidity = 0.07 uT.

Wood carbon showed higher percentages of CYN removal for the range of concentrations studied. The higher CYN adsorption capacity may be related to the higher volume of mesopores of wood carbon, which may have favoured the retention of the CYN molecule. Based on the molecular dimensions of the CYN, $0.911 \times 1.174 \times 1.218$ [54], the textural characteristics of carbon with a higher volume of mesopores (diameter between 2 and 50 nm) seem to be more suitable for adsorption of CYN molecule. Valencia-Cárdenas [20], Fonseca et al. [54] and Ho et al. [55] also found higher CYN adsorption capacities by activated carbons with high volumes of mesopores.

The adsorption equilibrium experimental data fit to the Freundlich, Langmuir and Redlich-Peterson isotherm models. The adjustment of the experimental data was performed by linear and non-linear regression, except for the Redlich-Peterson model in which the adjustment was accomplished only by non-linear regression since this model has three parameters.

Once the parameters of the models were obtained, the values of adsorptive capacity were calculated as a function of the equilibrium concentrations of CYN (Ce) obtained in the experiments. The parameters obtained and the experimental and calculated data are presented in Table 4 and Figure 8 show the results of experimental data adjustments to the isotherm models.

The experimental data of adsorption capacity as a function of CYN concentration at the equilibrium condition for Bituminous carbon showed an excellent fit to the Freundlich, Langmuir, and Redlich-Peterson models, indicated by the $R^2$ values greater than 0.94, for the adjustment by both linear and non-linear regression. The excellent experimental data fit the isothermal models were also evidenced by the low estimated normalized standard deviation ($\Delta q$).

The isotherm obtained in the adsorption equilibrium tests for bituminous carbon showed the best fit to the Freundlich model by linear regression with $R^2 = 0.978$, indicating that CYN adsorption occurs in multilayers in adsorption sites with different adsorption energies [56].

**Table 4.** Isotherm constants for the adsorption of CYN onto Bituminous and Wood carbons using Paranoá lake water as matrix.

| Isotherm Models | Equation | Parameters | Bituminous GAC | | Wood GAC | |
|---|---|---|---|---|---|---|
| | | | Linear | Non-Linear | Linear | Non-Linear |
| Freundlich | $q_e = K_F\, C_e^{1/n}$ | $1/n$ | 0.1993 | 0.2137 | 0.2905 | 0.3051 |
| | | $K_F\ (\mu g/mg)(L/\mu g)^{1/n}$ | 0.9935 | 0.9364 | 1.2354 | 1.2164 |
| | | $\Delta q(\%)$ | 3.196 | 4.782 | 19.4598 | 20.3861 |
| | | $R^2$ | 0.9783 | 0.9506 | 0.7813 | 0.7342 |
| Langmuir | $q_e = \dfrac{q_{max}\, K_L\, C_e}{1+K_L C_e}$ | $q_{max}\ (\mu g/mg)$ | 2.3047 | 2.4029 | 3.667 | 3.8610 |
| | | $K_L\ (L/\mu g)$ | 0.2542 | 0.2068 | 0.2791 | 0.2438 |
| | | RL Factor | 0.0370 | 0.0452 | 0.0386 | 0.0439 |
| | | $\Delta q(\%)$ | 5.2325 | 5.9092 | 9.229 | 9.8743 |
| | | $R^2$ | 0.957 | 0.941 | 0.9754 | 0.8992 |
| Redlich-Peterson | $q_e = \dfrac{K_R\, C_e}{1+a_R\, C_e^{\beta}}$ | $K_R\ (L/mg)$ | | 2.1193 | | 0.9411 |
| | | $a_R\ (L/\mu g)$ | | 1.7734 | | 0.2438 |
| | | $\beta$ | | 0.8425 | | 1.0000 |
| | | $\Delta q\ (\%)$ | | 3.0323 | | 9.8743 |
| | | $R^2$ | | 0.9778 | | 0.8992 |

$q_{max}$ maximum concentration of the adsorbate in the adsorbent per mass of adsorbent when the adsorption sites are saturated with the adsorbate, $K_L$: Langmuir adsorption constant, RL Factor: dimensionless constant, $K_F$: Freundlich constant, $1/n$: Freundlich adsorption intensity, $K_R$ e a R; Redlich-Peterson isotherm constant, $\beta$: exponent of the Redlich-Peterson isotherm.

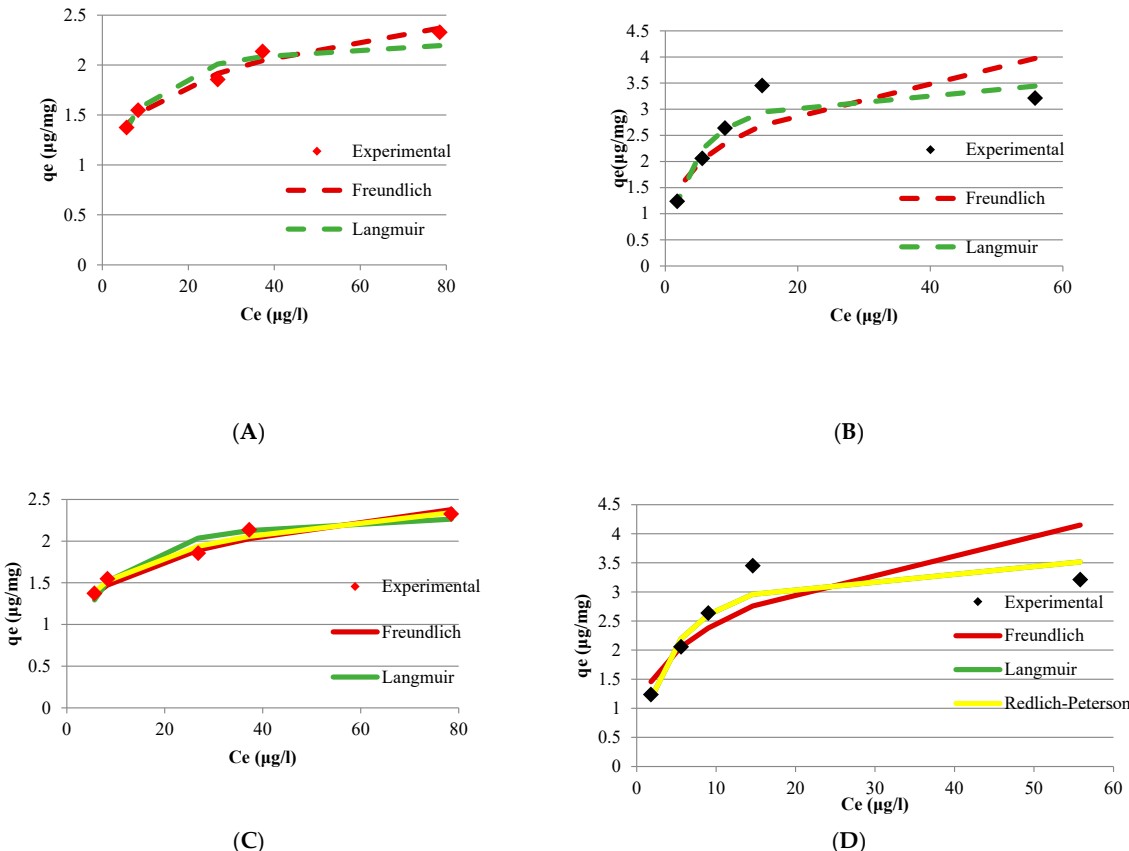

**Figure 8.** (**A**) Bituminous GAC, experimental and calculated data, parameters obtained by linear regression; (**B**) Wood GAC, experimental and calculated data, parameters obtained by linear regression; (**C**) Bituminous GAC, experimental and calculated data, parameters obtained by non-linear regression; (**D**) Wood GAC, experimental and calculated data, parameters obtained by non-linear regression.

On the other hand, the isotherm obtained in the adsorption equilibrium tests using Wood carbon showed a better fit to the Langmuir model, with $R^2$ values of 0.9754 and 0.9252, obtained by linear and non-linear regression, respectively. The best fit of the experimental data to the Langmuir model allows inferring that in wood carbon, the adsorption of CYN tends to occur in a monolayer in adsorption sites with equivalent energy and absence of interaction between the adsorbed molecules [57].

Tran et al. [58] have highlighted the limitations associated with fitting experimental data of the adsorption capacity of adsorbents to the linear forms of the Langmuir model. According to the authors, the transformation of data to the linear form of the model equation can lead to modifications of structure error, the introduction of error into the independent variable, and alteration of the weight placed on each data point. According to the authors such issues often lead to differences in the fitted parameter values between linear and non-linear forms of the Langmuir model. Nevertheless, under the experimental conditions evaluated in the present study, fitting experimental data to the linear form of the Langmuir model does not seem to significantly modify the values of the model's parameters compared to those obtained by fitting the experimental data to the non-linear form of the mathematical equation. Table 4 shows that the values of both parameters were very close for both carbons. For the wood carbon, with the higher adsorption capacity of CYN, the values determined by the linearized and non-linearized model equation, values were, respectively, 0.2791 L/µg and 0.2438 L/µg for $K_L$, and 3.667 µg/mg and 3.8610 µg/mg for $q_{max}$.

The essential characteristics of the Langmuir isotherm can be expressed by the dimensionless constant, called the RL factor, obtained from Equation (2):

$$RL = \frac{1}{1 + K_L C_O} \tag{2}$$

Depending on the RL values, adsorption can be considered favorable (0 < RL < 1), unfavorable (RL > 1), linear (RL = 1) or irreversible (RL = 0) [59]. For both carbons, the RL factor presented a value from 0 to 1, indicating that the CYN adsorption process was favorable in both adsorbents under the conditions studied.

The regression linear adjustment of Langmuir model revealed the wood carbon had a higher CYN adsorption capacity than the bituminous carbon, with a $q_{max}$ value of 3.6670 µg of CYN adsorbed per mg of carbon.

Experimental data also fitted nonlinear regression to the Redlich-Peterson model. This empirical model is composed of three parameters and result from the Langmuir and Freundlich models combination. Thus, this model includes a hybrid adsorption mechanism that does not follow the ideal adsorption in monolayer.

The isotherm obtained in the adsorption equilibrium tests for bituminous carbon showed an excellent fit to the Redlich-Peterson model with $R^2 = 0.977$. The satisfactory fit of the isotherm to this model corroborates the adsorption of CYN in this carbon occurs in multilayers.

Differently, the adjustment of the Redlich-Peterson model to the experimental data obtained for wood carbon resulted in a value of the parameter β = 1, which makes the mathematical expression of the Redlich-Peterson model equal to the that of the Langmuir model. Hence, in both models the $R^2$ values were similar for the adjustment by non-linear regression.

In the recent literature review, few studies evaluated the fitting of adsorption isotherm models to CYN removal by activated carbons Tables 5 and 6 present, respectively, the parameter values of the Freundlich and Langmuir isotherm models found in studies of CYN adsorption on activated carbons of different origins and surface characteristics. Studies applying the Redlich-Peterson model were not found.

**Table 5.** Freundlich parameters for CYN removal in activated carbon—Bibliographic references.

| Material | Area BET ($m^2/g$) | Conditions | $K_F$ ($\mu g/mg$)($L/\mu g$)$^{1/n}$ | $1/n$ | $M_o$ (mg) | References |
|---|---|---|---|---|---|---|
| Carbon | 1208 | Raw Water; pH = 7.9; Co = 37.4 $\mu g/L$ | 0.17 | 0.02 | 100 | [55] |
| Wood | 519 | Raw Water; pH = 7.9; Co = 37.4 $\mu g/L$ | 0.13 | 0.05 | 130.77 | [55] |
| Carbon Pre-Load | 1057 | Ultrapure Water; pH = 6.6; Co = 100 $\mu g/L$ | 0.12 | 0.69 | 141.67 | [60] |
| Carbon | 1057 | Ultrapure Water; pH = 6,6; Co = 100 $\mu g/L$ | 1.76 | 0.33 | 9.66 | [60] |
| Wood | 1813 | Ultrapure Water; pH = 6.6; Co = 100 $\mu g/L$ | 0.21 | 0.68 | 80.95 | [60] |
| Coconut | 1568 | Ultrapure Water; pH = 6.6; Co = 100 $\mu g/L$ | 1.34 | 0.38 | 12.69 | [60] |
| Bitumen | 819 | Ultrafiltered Water; pH = 6.5; Co = 102.23 $\mu g/L$ | 0.99 | 0.2 | 17.11 | Present study |
| Wood | 726 | Ultrafiltered Water; pH = 6.5; Co = 89.24 $\mu g/L$ | 1.24 | 0.29 | 13.76 | Present study |

**Table 6.** Langmuir parameters for CYN removal in activated carbon—Bibliographic references.

| Material | Area BET ($m^2/g$) | Conditions | $q_{max}$ ($\mu g/mg$) | KL ($L/\mu g$) | RL Factor | $M_o$ (mg) | References |
|---|---|---|---|---|---|---|---|
| Wast tyre | Uninformed | Ultrapure Water—pH = 3; Co = 65 $\mu g/L$ | 0.11 | 10.10 | 0.0015 | 169.85 | [61] |
| Bitumen | 819 | Ultrafiltered Water; pH = 6.5; Co = 102.23 $\mu g/L$ | 2.30 | 0.25 | 0.0371 | 36.40 | Present study |
| Wood | 726 | Ultrafiltered Water; pH = 6.5; Co = 89.24 $\mu g/L$ | 3.67 | 0.28 | 0.0386 | 21.25 | Present study |

Where $M_o$: Mass of carbon required to achieve Ce = 1 $\mu g/L$ considering: Co = 35 $\mu g/L$ and the volume of water = 0.5 L.

According to Table 5, the material with the highest CYN adsorption capacity, and indicated by the highest value of the $K_f$ parameter, was a carbon-based adsorbent [60]. However, the same adsorbent presented the lowest CYN adsorption capacity when exposed to water treated by sedimentation before the adsorption process (Pre-Loading). The reduced adsorption capacity is likely associated with the organic matter present in the preloading carbon that may have directly occupied the adsorption sites during the preloading process [60].

The carbons used herein seemed to have the highest adsorption capacities among the carbons evaluated in a matrix with the presence of organic matter that competes with the toxin for the adsorption sites. Although the NOM concentration was not quantified, likely the study water contained NOM from the Paranoá Lake ultrafiltered water besides that CYN extract algogenic organic matter. Even under competition conditions with organic matter, the carbons evaluated in this study showed higher CYN adsorption capacity when compared to the adsorption capacity of wood-based carbon [60] in ultrapure water matrix. This superiority, evidenced by the higher $K_f$ coefficients and lower carbon mass, allows a residual concentration of CYN in the treated water of 1 $\mu g/L$.

Only one study using the Langmuir model was found in the literature, as shown in Table 6. The waste tire-based carbon has a lower CYN adsorption capacity than the carbons tested herein, evidenced by the higher mass of carbon required to obtain a residual CYN

concentration of 1 μg/L. Although tire-based carbon had a lower adsorptive capacity, the value of the KL parameter indicates that the energy involved in the CYN adsorption on this adsorbent is higher than the energy for the other carbons. The value of the RL factor closer to zero corroborates the higher energy involved, pointing out the irreversibility of the adsorptive process.

The adsorption equilibrium tests showed the wood-activated carbon as the adsorbent with the highest CYN adsorption capacity. Therefore, the subsequent experimental stage, SBA tests and mathematical modeling, was performed to investigate the performance of this carbon to remove CYN in a fixed bed GAC absorber column along with the treatment of ultrafiltrated Paranoá Lake water.

### 3.4. SBA Tests

Figure 9 shows the results of SBA tests performed in duplicate.

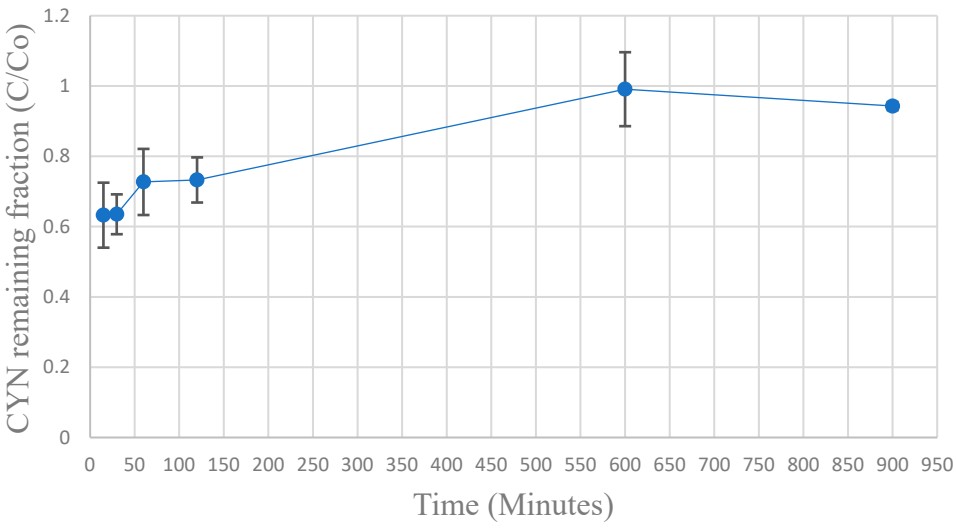

**Figure 9.** CYN breakthrough curve obtained in SBA tests—Average values of CYN remaining fraction in the effluent. The bars represent the standard deviations of average values.

As recommended by Weber and Liu [39], the column starts its operation with an effluent concentration of CYN different from zero, which increases during the test.

Using the average values of the CYN remaining fraction of effluent ($C/C_0$) obtained in the SBA tests, a simulation of the adsorbing column of GAC in operation was performed using the HSDM model with the aid of the FAST 2.1 software [52].

The input data applied to the HSDM model were:

- adsorption coefficients of Langmuir isotherm ($q_{max}$ and KL), obtained from the fitting of experimental data to the isotherm model;
- the average influent concentration of CYN and the mass of GAC, obtained using the SBA tests.

The parameters of the Wood activated carbon used in the adsorbing column: apparent bed density, average particle diameter and axial velocity, were the same used in the SBA tests.

The mass transfer coefficients $K_f$ (film diffusion coefficient) and $D_s$ (surface diffusion coefficient) were obtained by fitting the experimental data to the HSDM model using the least-squares method. The average value of the CYN remaining fraction in the effluent of the SBA column—$C/C_0$) fitted satisfactorily to the HSDM model with a coefficient of determination ($R^2$) of 0.89. Figure 10 displays the CYN breakthrough curve obtained from the simulation accomplished with the HSDM model.

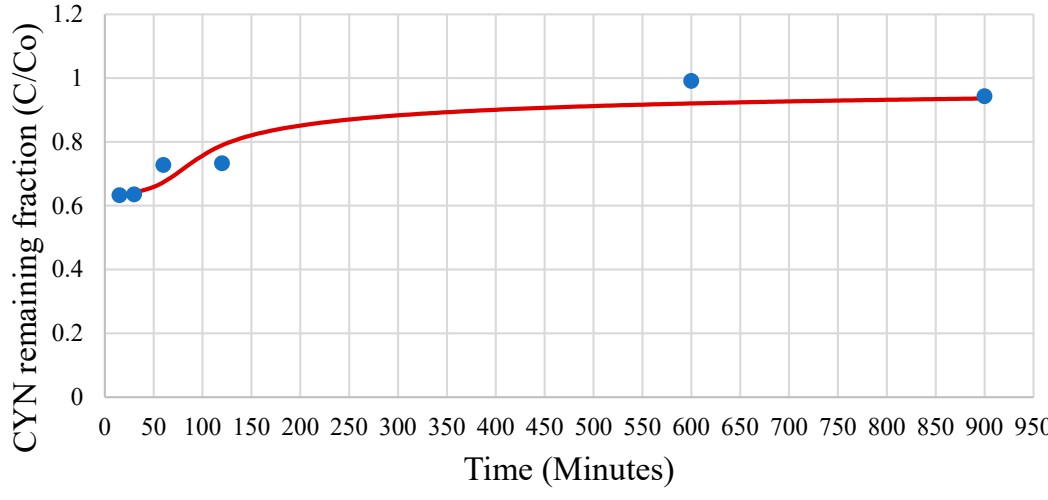

**Figure 10.** Average CYN remaining fraction (C/C$_0$) in the effluent from the SBA column and values predicted by the HSDM model. Input parameters used in the model: Mass of GAC = 1.1296 g; Apparent bed density = 0.476 g/cm$^3$; Grain density = 0.850 g/cm$^3$; Average particle diameter = 0.855 mm; Initial CYN concentration = 95.5 µg/L; axial velocity = 200 m$^3$/(m$^2$.day); Equilibrium coefficients (Langmuir isotherm) q$_{max}$ = 3670 µg/g, K$_L$ = 0.2791 L/µg; Mass transfer coefficients D$_s$ = 3 × 10$^{-16}$ m$^2$/s and K$_f$ = 9 × 10$^{-6}$ m/s.

The calibration of the HSDM model using the values of the mass transfer coefficients made it possible for the model to satisfactorily simulate the operating dynamics of the GAC column used in the SBA tests. The model also accurately reproduced the experimental CYN remaining fraction during the initial and final phases of the SBA test.

A sensitivity analysis of the HSDM model was conducted using the values of the mass transfer coefficients to verify the changes in the mass transfer coefficients in conditions the model is simulating the operation of a full-scale GAC column. Initially, the HSDM model simulated the operation of the adsorber column with the mass transfer coefficients that allowed the best fit in the model calibration (D$_s$ = 3 × 10$^{-16}$ m$^2$/s e K$_f$ = 9 × 10$^{-6}$ m/s). Then simulations were performed keeping the value of D$_s$ = 3 × 10$^{-16}$ m$^2$/s and changing K$_f$ to values 50% higher and lower than the best fit value of this parameter. Later, HSDM model simulated changing the D$_s$ to values 50% higher and lower in relation to the best fit value of D$_s$ keeping the value of K$_f$ = 9 × 10$^{-6}$ m/s. Figures 11 and 12 show the results of HSDM model simulations.

When simulating the operation of the column under full scale conditions with the mass transfer coefficients that best fit the model (Figure 11), the column reaches the breaking point (CYN effluent concentration = 1 µg/L) in 182 h. Additionally, 50% reduction in the K$_f$ value reduces the breakthrough time by 33%. On the other hand, increasing K$_f$ by 50% results in an increase in breakthrough time of 11%.

Concerning D$_s$, a 50% reduction of this coefficient to the value fitted to the model (Figure 12) reduced the breakthrough time by 49%. In contrast, a 50% increase in the coefficient promoted a 45% increase in the breakthrough time, evidencing that the coefficient D$_s$ had a more significant influence on the breakthrough curve's behavior.

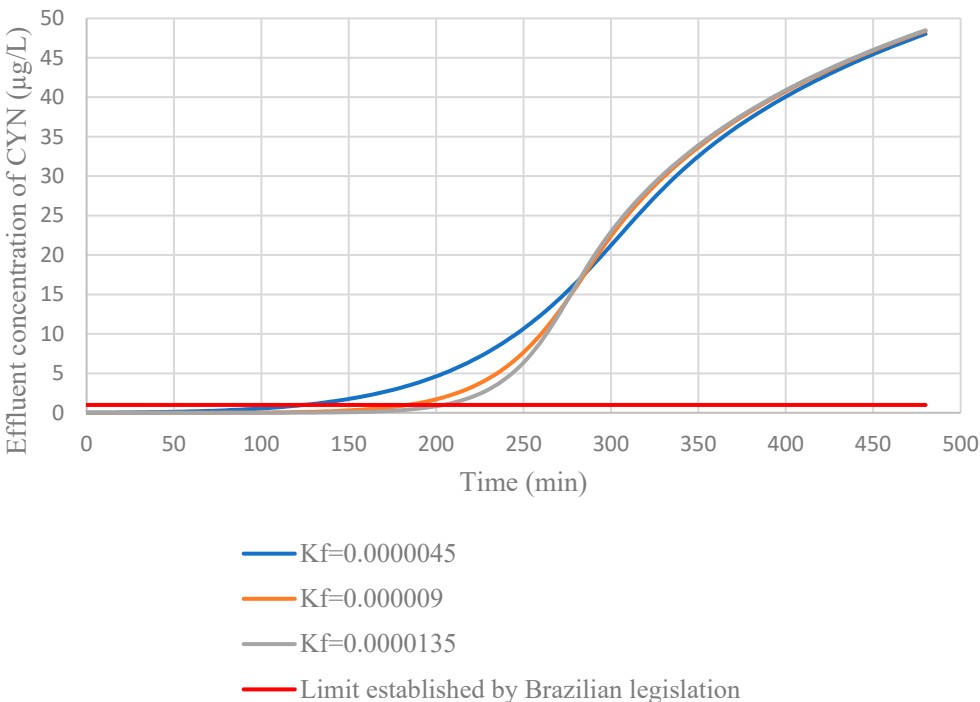

**Figure 11.** Sensitivity analysis of the model from changes in the value of the mass transfer coefficient, Kf. Input values used in the model: EBCT = 10 min; Apparent bed density = 0.476 g/cm$^3$; Grain density = 0.850 g/cm$^3$; Average grain diameter = 0.855 mm; Initial CYN concentration = 100 µg/L; Flow = 700 L/s; Equilibrium coefficients (Langmuir isotherm) $q_{max}$ = 3670 µg/g, KL = 0.2791 L/µg; Surface diffusion coefficient, $D_s$ = 3 × 10$^{-16}$ m$^2$/s.

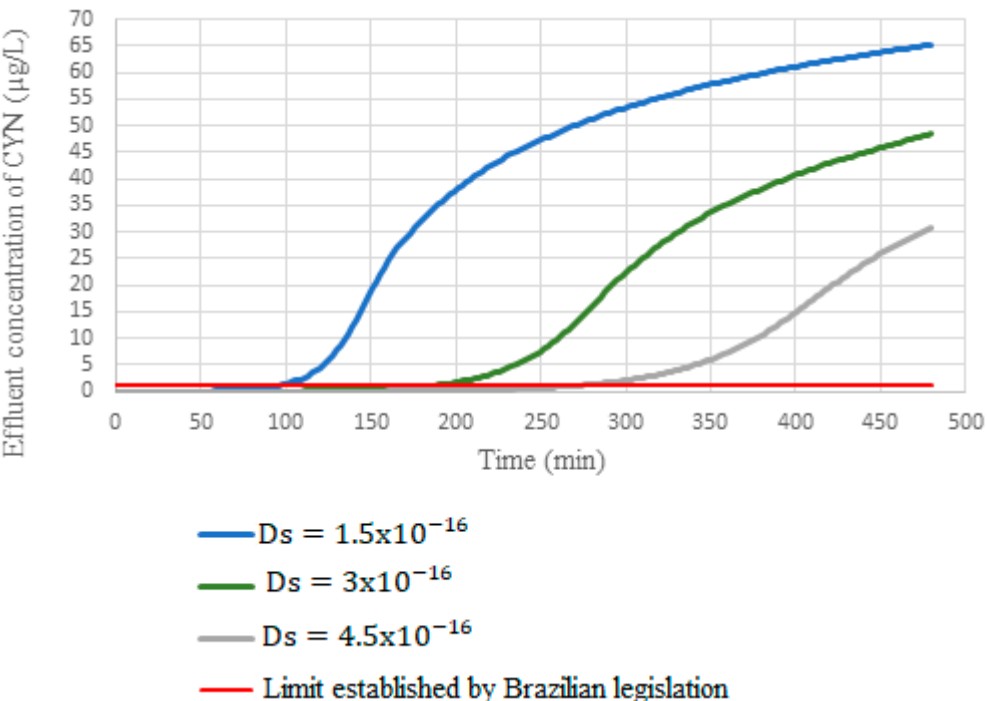

**Figure 12.** Sensitivity analysis of the model against changes in the values of the mass transfer coefficient, Ds. Input values used in the model: EBCT = 10 min; Apparent bed density = 0.476 g/cm$^3$; Grain density = 0.850 g/cm$^3$; Average grain diameter = 0.855 mm; Initial CYN concentration = 100 µg/L; Flow = 700 L/s; Equilibrium coefficients (Langmuir isotherm) $q_{max}$ = 3670 µg/g, KL = 0.2791 L/µg; Film diffusion coefficient, $K_f$ = 9 × 10$^{-6}$ m/s.

These results demonstrate that $K_f$ values changes seem to influence only the initial stages of adsorbing column operation. On the other hand, changes in the $D_s$ coefficient tend to influence the behavior of the breakthrough curve throughout the entire operating time, corroborating the study by Cook and Newcombe [62], which consider $D_s$ the main fitting parameter of the HSDM model. However, for the analysis of the initial stages of the breakthrough curve, it is important to accurately determine the two mass transfer coefficients since both influenced on the initial stages of the breakthrough curve.

Full-scale simulations of GAC adsorbing column operation were also performed by changing the empty bed contact times. Figure 13 shows the results.

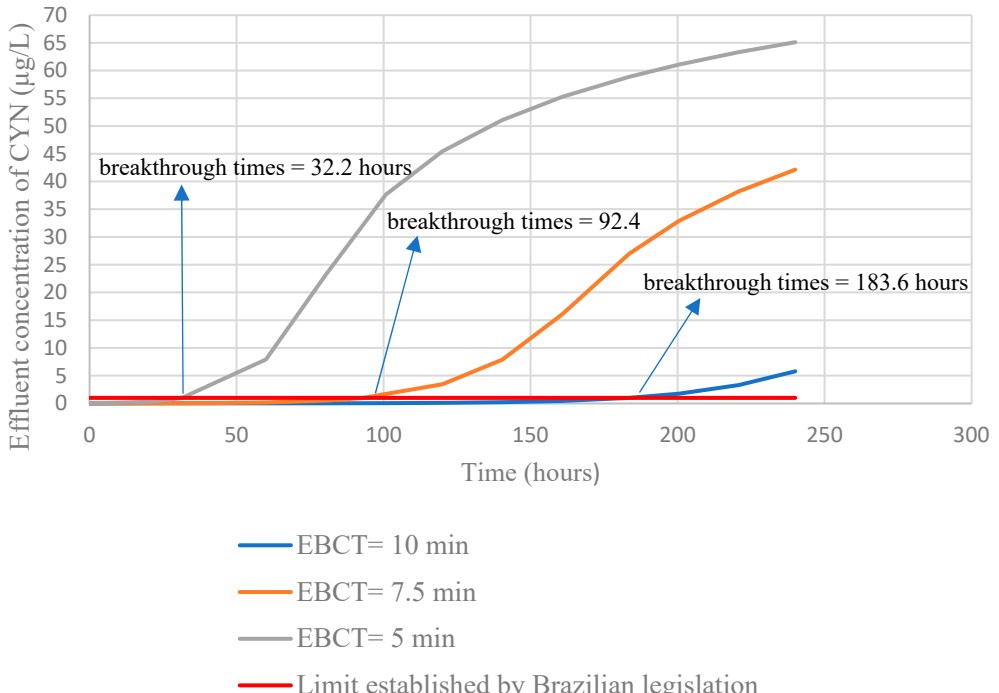

**Figure 13.** Simulation of the full-scale GAC adsorbent column operating with different EBCTs. Input values used in the model: Apparent bed density = 0.476 g/cm$^3$; Grain density = 0.850 g/cm$^3$; Average grain diameter = 0.855 mm; Initial CYN concentration = 100 µg/L; Flow = 700 L/s; Equilibrium coefficients (Langmuir isotherm) $q_{max}$ = 3670 µg/g, KL = 0.2791 L/µg; Mass transfer coefficients $D_s = 3 \times 10^{-16}$ m$^2$/s e $K_f = 9 \times 10^{-6}$ m/s.

The simulations indicated that the increase in the EBCT promotes an increase in breakthrough times. Corwin and Summers [63] observed a similar trend in breakthrough time when studying the removal of organic matter in GAC columns by employing rapid tests in a reduced scale column. On the other hand, the authors verified an inverse trend when analyzing the removal of Bisphenol-A when the residual ($C/C_0$) in the effluent was higher than 0.1. The authors explained this behavior considering that at the beginning of the column operation ($C/C_0 < 0.1$) the mass transfer zones of organic matter and Bisphenol-A overlap, and the carbon adsorption sites are available for both contaminants. Nevertheless, throughout the column operation time, the mass transfer zone of organic matter advances faster than that of Bisphenol-A, with the adsorption sites being occupied by the organic matter, thus preventing the subsequent adsorption of Bisphenol-A.

In the present study, both the adsorption equilibrium coefficients and the mass transfer coefficients were obtained in experiments carried out with study water containing organic matter, thus indirectly considering the influence of the competition between organic matter and CYN in the simulations by the HSDM model.

Finally, Table 7 shows the carbon use rate using the estimated breakthrough for different empty bed contact times.

**Table 7.** Carbon use rate for different empty bed contact time values.

| Empty Bed Contact Times (min) | Carbon Use Rate (Kg/m$^3$) |
| --- | --- |
| 5.0 | 1.27 |
| 7.5 | 0.64 |
| 10.0 | 0.43 |

An increase in the EBCT results in a reduction in the carbon use rate, which characterizes a more efficient use of GAC and, consequently, reduced costs associated with the operation of the GAC column, which is relevant information for the design of adsorbing columns of GAC in studies on CYN removal.

### 4. Conclusions

In the present study, two commercial activated carbons: Bituminous carbon and Wood carbon, were investigated, at bench scale, to their capacity of adsorbing cylindrospermopsin, aiming its application in granular form in fixed bed adsorber columns.

The textural characterization of the carbons pointed out that the wood carbon is richer in mesopores (BET surface area of 726 m$^2$/g) than bituminous carbon, with more microporous structure associated with a larger BET surface area (819 m$^2$/g).

The adsorption isotherms revealed the wood carbon, exhibited a higher cylindrospermopsin adsorption capacity ($q_{max}$ = 3.67 µg/mg) than bituminous carbon ($q_{max}$ = 2.30 µg/mg). The Langmuir, Freundlich and Rendlich-Peterson models accurately described the adsorption of cylindrospermopsin by bituminous carbon. On the other hand, the Langmuir model better represented the adsorption of cyanotoxin by wood carbon with R$^2$ of 0.9754, for the adjustment by linear regression.

Based on the behavior of the isotherms, Wood carbon, was chosen to develop the kinetic tests in short bed adsorber columns (SBA) and to evaluate the behavior of the breakthrough curve using the HSDM model. The HSDM model satisfactorily simulated (R$^2$ = 0.89) the dynamics of operation of the GAC column in the SBA tests, accurately reproducing the cylindrospermopsin breakthrough curve to the film diffusion coefficient ($K_f$) of $9 \times 10^{-6}$ m/s and surface diffusion coefficient ($D_s$) of $3 \times 10^{-16}$ m$^2$/s.

Sensitivity analysis of the HSDM model showed that changes in the values of the film diffusion coefficient ($K_f$) tend to influence the initial stages of column operation and that changes in the surface diffusion coefficient ($D_s$) influence the behavior of the breakthroughcurve throughout the operating time.

The breakthrough times estimated by the HSDM Model for cylindrospermopsin for different empty bed contact time (EBCT) conditions indicated that the carbon use rate decreases with the increase of EBCT, leading to a lower cost of full-scale GAC column operation. The best carbon use rate found was 0.43 kg/m$^3$ for a EBCT of 10 min and breakthrough time of 183.6 h.

Despite published studies concerning CYN adsorption onto activated carbons to the best of our knowledge, this is the first study evaluating the cylindrospermopsin removal in fixed-bed adsorber columns. Therefore, it is essential to conduct additional studies using other tools and tests, such as RSSCT and pilot-scale studies, to acquire more insight into GAC absorber columns' design and operation parameters aiming its application in full-scale water treatment systems.

Finally, it is also necessary to assess the influence of different concentrations of CYN and organic matter on the CYN breakthrough curve as their levels tend to vary considerably in aquatic environments.

**Author Contributions:** Conceptualization, C.C.A.; methodology, C.C.A.; validation, C.C.A. and Y.P.G.; formal analysis, C.C.A. and Y.P.G.; investigation, C.C.A.; data curation, C.C.A. and Y.P.G.; writing—original draft preparation, C.C.A.; writing—review and editing, Y.P.G.; visualization, Y.P.G.; supervision, Y.P.G.; project administration, Y.P.G.; funding acquisition, Y.P.G. All authors have read and agreed to the published version of the manuscript.

**Funding:** This research was funded by CNPq, grant number [133239/2019-5].

**Acknowledgments:** We gratefully acknowledge FAP-DF, CNPq and FUNASA for their financial Support and CAESB for kindly providing the base water. We also thank staff of the Environmental Sanitation Laboratory at the University of Brasília for their helpful cooperation. The authors gratefully acknowledge the editor and anonymous reviewers for their invaluable insight and helpful suggestion.

**Conflicts of Interest:** The authors declare no conflict of interest.

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
