# Peer review of "Removal of Cylindrospermopsin by Adsorption on Granular Activated Carbon, Selection of Carbons and Estimated Fixed-Bed Breakthrough"

_water, doi:10.3390/w14101630_

Round 1

Reviewer 1 Report

Please see my comments in the word file.

Reviewer 2 Report

The paper reported the removal of cyanotoxins with carbon materials. The topic is interesting to the journal readership. I recommended a major revision, and the related comments are as follows.

  1. The abstract should be revised to be concise. The important results and key parameters are not found here.
  2. The introduction was too long to be read effectively.
  3. If possible, the figures 6 and 7 can be combined, and then the kinetics models can be used here.
  4. Fig.8, the x axis is difficult to be followed. Carbon concentration, did you mean “carbon dosage”?
  5. Many figures should be revised in accordance with the journal requirement.
  6. If possible, a model should be used for the column experiment.
  7. The adsorption mechanism should be deeply discussed, along with an illustration graph.

Round 2

Reviewer 1 Report

Please see my comments in the file.

Reviewer 2 Report

The revised draft was in agreement with the reviewers suggetions, so that I recommended an acceptance. 

Author Response

Dear Reviewer,

Enclosed you will find the revised version of our manuscript, “Removal of cylindrospermopsin by adsorption on granular activated carbon, selection of carbons and estimated fixed-bed breakthrough” which were submitted to the Journal Water Special Issue " Harmful Cyanobacterial Blooms (HCBs) in Freshwaters–an Increasing Global Concern". The authors would like to thank the Reviewer for all their comments.